# From Traditional Ethnopharmacology to Modern Natural Drug Discovery: A Methodology Discussion and Specific Examples

**DOI:** 10.3390/molecules27134060

**Published:** 2022-06-24

**Authors:** Stergios Pirintsos, Athanasios Panagiotopoulos, Michalis Bariotakis, Vangelis Daskalakis, Christos Lionis, George Sourvinos, Ioannis Karakasiliotis, Marilena Kampa, Elias Castanas

**Affiliations:** 1Department of Biology, School of Sciences and Technology, University of Crete, 71409 Heraklion, Greece; navaak@gmail.com; 2Botanical Garden, University of Crete, 74100 Rethymnon, Greece; 3Nature Crete Pharmaceuticals, 71305 Heraklion, Greece; lionis@uoc.gr (C.L.); sourving@uoc.gr (G.S.); kampam@uoc.gr (M.K.); 4Laboratory of Experimental Endocrinology, School of Medicine, University of Crete, 71409 Heraklion, Greece; athpanagiotopoulos@hotmail.com; 5Department of Chemical Engineering, Cyprus University of Technology, Limassol 3603, Cyprus; evangelos.daskalakis@cut.ac.cy; 6Clinic of Social and Family Medicine, School of Medicine, University of Crete, 71409 Heraklion, Greece; 7Laboratory of Clinical Virology, School of Medicine, University of Crete, 71409 Heraklion, Greece; 8Laboratory of Biology, School of Medicine, Democritus University of Thrace, 68100 Alexandroupolis, Greece; ioakarak@med.duth.gr

**Keywords:** ethnopharmacology, drug discovery, computational chemistry, bioprospecting, plants, experimental screening, in silico screening, pharmacological testing

## Abstract

Ethnopharmacology, through the description of the beneficial effects of plants, has provided an early framework for the therapeutic use of natural compounds. Natural products, either in their native form or after crude extraction of their active ingredients, have long been used by different populations and explored as invaluable sources for drug design. The transition from traditional ethnopharmacology to drug discovery has followed a straightforward path, assisted by the evolution of isolation and characterization methods, the increase in computational power, and the development of specific chemoinformatic methods. The deriving extensive exploitation of the natural product chemical space has led to the discovery of novel compounds with pharmaceutical properties, although this was not followed by an analogous increase in novel drugs. In this work, we discuss the evolution of ideas and methods, from traditional ethnopharmacology to in silico drug discovery, applied to natural products. We point out that, in the past, the starting point was the plant itself, identified by sustained ethnopharmacological research, with the active compound deriving after extensive analysis and testing. In contrast, in recent years, the active substance has been pinpointed by computational methods (in silico docking and molecular dynamics, network pharmacology), followed by the identification of the plant(s) containing the active ingredient, identified by existing or putative ethnopharmacological information. We further stress the potential pitfalls of recent in silico methods and discuss the absolute need for in vitro and in vivo validation as an absolute requirement. Finally, we present our contribution to natural products’ drug discovery by discussing specific examples, applying the whole continuum of this rapidly evolving field. In detail, we report the isolation of novel antiviral compounds, based on natural products active against influenza and SARS-CoV-2 and novel substances active on a specific GPCR, OXER1.

## 1. Introduction

Humans possess discrete pharmacological knowledge of the therapeutic properties of plants from the beginning of their evolutionary history, leaving imprints in prehistoric and later cultural heritage [1,2]. However, this knowledge, accumulated in traditional medicine and ethnopharmacology, is declining. Nevertheless, elements of ethnopharmacological knowledge and practice, exercised either in parallel or supplementing the official treatment of diseases, create significant pressure on the scientific community to provide data related to the safety and effectiveness of the natural extracts. This element remains partially unfulfilled until now.

The use of plant extracts, containing hundreds of chemicals as pharmaceutical agents, is no longer a black box or the primary obstacle to understanding their mechanisms of action and/or the contained “active compound(s)”. This is powered by the significant increase in the detection and precision limits of analytical methods, the significant multiplication in computational power, and the construction of large public libraries of chemical (and natural) compounds (Figure 1, upper part). Indeed, many circulating drugs derive from natural products, and many emblematic drugs, such as taxol [3,4,5], vinblastine [6,7], quinine [8,9], and artemisinin [10], are rooted in traditional medicine and ethnopharmacology [11]; nevertheless, at a later stage, these drugs have been synthesized by modern chemistry and re-evaluated with modern analytical and pharmacological methods. Taxol (known as paclitaxel) is a nitrogen-containing diterpenoid isolated from the bark of *Taxus brevifolia* Nutt., which acts as a tubulin stabilizer and leads to cell cycle arrest, acting as an anticancer agent. Vinblastine and vincristine are closely related indole dihydroindole dimers (bisindole alkaloids), isolated from *Catharanthus roseus* (L.) G. Don (formerly known as *Vinca rosea* L.), the Madagascar periwinkle. Both of these anticancer agents, known as vinca alkaloids in the medical literature, are specific binders of tubulin, leading to tubulin depolymerization and cell cycle arrest in the metaphase stage. Quinine is an alkaloid obtained from *Cinchona* spp. It was the first antimalarial drug and served as an effective remedy for this potentially lethal infectious disease in colonial times, making possible European settlement in many tropical and subtropical parts of the world. Finally, artemisinin is a sesquiterpene lactone antimalarial compound with an endoperoxide group, discovered as a constituent of *Artemisia annua* L., with a unique mechanism of action on the heme complex.

In recent years, innovative extraction technologies, including semi-bionic extraction; supercritical fluid extraction; microwave-assisted, ultrasonic-assisted, and enzyme-assisted extraction; molecular distillation methods; membrane separation technology; and sophisticated new methodologies and instrumentation such as HPLC-MS, LC-MS, GC-MS, NMR, and crystallography, in parallel with the development of biology and clinical and experimental medicine, have allowed the re-evaluation of the corpus of traditional knowledge, the determination of chemical components of plant extracts, the identification of “active compound(s)”, and the development of novel drugs [12,13]. Galantamine [14], an Amaryllidaceae-type alkaloid from *Galanthus woronowii* Losinsk and other species of this genus, which has been recently approved for the treatment of early-onset Alzheimer’s disease, is an example of a recent achievement. A detailed presentation of plant-derived drug discovery in the last 30 years has been extensively reviewed and discussed in a recent series of publications [13,15,16,17].

Plants synthesize an immensely rich diversity of specialized secondary metabolites comprising an enormous number of active or complementary compounds [18,19]. This is due to several reasons, including (1) the high plant biodiversity in many areas of the world; (2) their significant ecological role in plant physiology, which is related to the high variability of problems that the plants have to face (protection from herbivores, pathogens, stress (including UV protection), other plant–plant and plant–animal interactions, etc); (3) the fact that, for the same problem, different evolutionary solutions have appeared in divergent plant lineages, with identical or similar pharmacological action [18,20]; and (4) the fact that different parts of the plant and different extraction methods of the same plant may result in a different collection of active compounds, with sometimes opposing biological effects [21]. Therefore, the validating process of ethnopharmacological knowledge is a laborious, and usually partially successful, enterprise [11], taking into account that only 1/10,000 tested compounds may lead to a successful drug in a time frame of almost ten years [22].

The exponential increase in computational power and data storage capabilities in recent decades has led to faster and, in some cases, economically sustainable solutions for drug discovery. The development of chemical libraries with billions of compounds and specific libraries of existing or putative natural compounds, with hundreds or thousands of molecules, together with the development of novel computational approaches (in silico docking methods, assisted by molecular dynamics, quantitative structure–activity relationships (QSARs), in silico evaluation of absorption–distribution–metabolism (ADMET), etc.), have been advanced as promising methods for the initial screening of the natural compound chemical space for a given disease (Figure 1, lower part) [23,24]. Computational high-throughput virtual screening has advanced as a cost-effective and less time-consuming method for drug discovery [22], as compounds from different chemical libraries have been subjected to high-throughput screening against a valid or presumed pathophysiological disease-related target. The first success of this approach was obtained in 1990, with the discovery of a dopamine D2 agonist [25]. Since then, the computational approach and virtual screening have been combined with network pharmacology (construction of signaling and interacting cellular networks, based on the observed or deducted interaction of compounds with cellular mechanisms). This has helped and accelerated drug discovery and development, positioning network pharmacology [26] as a paradigm shift in a newly emerged methodology, targeting all critical networks involved or perturbed in a disease. This approach complements the genomic, genetic, gene-related, and pathophysiological approach to disease. However, although computational chemistry has revolutionized the process of drug discovery, some limitations still exist, inherent to the accuracy of the computer programs used and the possible overfitting, induced by in silico methods, necessitating proper experimental validation.

Here, we discuss in brief the different steps in drug discovery, from ethnopharmacological observation to modern, high-throughput virtual screening, in an attempt to follow the rapid evolution of ideas in the field. Although a detailed review of all these topics is out of the scope of the current work, we review and discuss the progress and experimental approaches, applied to natural products, as depicted in Figure 1. We point out that in the past, the starting point was the plant itself, highlighted by sustained ethnopharmacological research, with the active compound deriving after extensive analysis and testing. In contrast, in the current state of scientific knowledge, the active substance is pinpointed by computational methods (in silico docking and molecular dynamics, quantitative structure–activity relationship (QSAR), network pharmacology, ADMET), followed by the identification of the plant(s) containing the active ingredient, seeking existing or possible ethnopharmacological information and relationships. We further stress the potential pitfalls of recent in silico methods and discuss the absolute need for in vitro and in vivo verification as a future recommendation. In addition, we present our contribution to this process, through specific examples of novel drug discovery. We further stress the potential pitfalls of recent in silico methods and discuss the absolute need for in vitro and in vivo verification of computer-generated data.

## 2. Evolution of Natural-Product-Derived Drug Development

### 2.1. Traditional Ethnopharmacology and Ethnobotany

Ethnopharmacology and ethnobotany are close neighboring fields. Ethnobotany is the study of complex relationships between cultures and their use of plants, focusing primarily on how plants are managed, used, and perceived across human societies. Ethnopharmacology, on the other hand, is defined as the interdisciplinary scientific exploration of traditionally employed indigenous drugs and biologically active agents [18,27,28,29]. Therefore, ethnopharmacology has a broader focus on exploring biologically active agents from plants, minerals, animals, fungi, and microbes. In both fields, a first step consists in the presentation of the use of extracts in a given disease, without investigating any potential causal relationship with contained ingredients/compounds (for a concrete example, see [30]). Ethnopharmacology has significantly contributed to the field explorations of indigenous and traditional medical knowledge and the biodiversity component to which such knowledge is linked.

In modern societies, this traditional use of plants as alternative pharmacological agents still persists. In China, traditional Chinese medicine is still serving many of the health needs of the population. It is practiced in parallel with modern medical treatment, due to the extensive recording of the plants’ medicinal properties. A series of institutions have been established to promote traditional Chinese medicine, such as the Academy of Traditional Chinese Medicine and training institutions. Almost all hospitals have an additional department of traditional medicine. Interestingly, the Chinese government proposes that both traditional Chinese and Western medicine can be combined to treat pneumonia caused by SARS-CoV-2, with promising results [31].

In contrast, western societies have mostly lost their traditional healing practice, although some elements of the plants’ medicinal properties have survived. Phytotherapists or naturopaths, operating within alternative and complementary health care systems, promote the ethnopharmacological use of plants. In some countries, they undergo training that is more or less regulated (in many cases by competent authorities) and have associations that recognize qualified members [32], making herbal medicine safe, effective, and standardized. In contrast, some other countries simply promote the exploitation of medicinal plants, without incorporating any regulatory policies, leading to the rapid loss of traditional practices. In the latter countries, the practice of ethnopharmacology and ethnobotany consists in the use of entire plant or crude plant extracts. It is worth mentioning, however, that the use of an entire plant, crude extract, or mixture of different plant extracts, with no isolation of components, results, in some cases, in a better therapeutic effect than the administration of individual compounds [33]. This is attributed to a synergy of active compounds included in the preparation (see [34] for a discussion and references therein). For this reason, teabags with the plant’s dried components, and bulk dried plant material both suggested as concoctions in use, are provided in the herbal markets [35].

Taking into account the above-discussed elements, an unbalanced perception of ethnopharmacology emerges, as this practice has the potential to contribute to the improvement of the health of indigenous people, supporting health care providers in the developing world, in addition to accelerating drug discovery [29,36]. In this respect, ethnopharmacology has, in our opinion, a characteristic of modernity, which ensembles particular sociocultural norms, attitudes, and practices.

Safety issues of herbal medicines is a global priority for national health authorities and the general public, both in their traditional use and in drug discovery [37,38]. At the European level, the Committee on Herbal Medicinal Products (HMPC), depending on the European Medicines Agency (EMA), is responsible for preparing the Agency’s opinions on the safety of herbal medicines. According to EMA regulations, clinical studies and tests on the safety and efficacy of herbal medicines are not required for their traditional use (local or per os administration), as long as this traditional use is well documented for at least 30 years, including at least 15 years within the EU (Directive 2004/24/EC of the European Parliament and of the Council of 31 March 2004).

In our contribution to the safety of herbal medicine [39], using partial-order analysis ([40], and the literature cited therein), we explored the reported undesirable effects of medicinal herbal substances, in their traditional/ethnopharmacological use. Data were retrieved from the European Union list of herbal substances for use in traditional medicinal products and the final European Union herbal monographs of the European Medicines Agency. Our analysis revealed that the reported undesirable effects were (in diminishing order) gastrointestinal disorders, allergic skin reactions, and allergic and hypersensitivity reactions. Variations in undesirable effects of herbal substances between plants of major phylogenetic groups of origin were also recorded, and an overall arrangement of medicinal herbal substances in rank order was obtained. This classification was proposed as a guide for the decision-making process for both healthcare providers and consumers. Moreover, taking into consideration that several data matrices have been published in many world areas, ranging from regional to continental scale, we attempted to demonstrate, for the first time, the implementation of partial-order techniques, processing ethnopharmacological information, with the purpose to reveal hidden inner structures and characteristics of reported raw data [41]. This methodology could potentially contribute to the conceptualization and management of ethnopharmacological knowledge [18].

### 2.2. Pharmacological Testing

Yeung et al. [42] analyzed the ethnopharmacology literature with regard to publication and citation data. They showed that research on recording medicinal plant species used by traditional medicine persists, but the evaluation of specific properties or treatment effects of extracts and compounds has increased enormously. Interestingly, the publications’ impact was directly related to the number of indigenous species in the authors’ countries. Currently, the trend of research has shifted from identifying and recording the medicinal plant species used in traditional medicine [30,43,44,45,46,47,48,49,50,51,52,53,54] to the evaluation of specific properties or treatment effects of crude plant extracts [55,56,57,58,59,60,61,62,63], or particular naturally derived products, such as flavonoids [64], alkaloids [65], tannins [66], saponins [67], phenols [68], and terpenoids [69] (an analysis of the provided bibliography together with applied methodology is presented in Table 1).

However, the explosion of studies in pharmacological testing of plant extracts and compounds did not translate into an analogous increase in drug production. This was due to the fact that the majority of studies were mainly descriptive and did not integrate the next necessary step, which is the implementation of a state-of-the-art clinical trial. Millions of molecules are tested, and thousands have been produced, but most of them fail to progress in preclinical or clinical settings [70], mainly due to the lack of clinical studies in the field [42]. Indeed, the first edge of the translation chain, leading from a plant extract to a final product, relies on the choice of plants, as well as the choice of secondary metabolites, whose ethnopharmacological history is expected to ensure the success of the pharmacological testing and the desired health benefit (drug design). According to Pirintsos et al. [34], even the supply of plant raw material may become an obstacle influencing the possibility of new drug production, as several restrictions rule the natural collections (harvests) and trade of herbs and spices, especially within the framework of the EU environmental policy, as well as within the framework of the United Nations Convention on Biological Diversity. Therefore, in order to translate the results of the pharmacological testing to a new drug product, and in order to overcome the “valley of death”, a necessity of the bridging between successional steps, or successional links of the translational chain, should always be taken into consideration [34].

In our contribution to pharmacological testing, Lionis et al. [71] revealed low morbidity and mortality rates of coronary heart disease in Crete and the existence of an indigenous knowledge system in rural Crete with certain combinations of different aromatic plants, which have been used for the prevention and cure of the common cold and influenza. *Origanum dictamnus* (wild and cultivated dittany), *Matricaria recutita (chamomile), Satureja thymbra (savory), Coridothymus capitatus (thyme), Mentha pulegium (penny royal), Salvia pomifera* (wild sage), *Salvia fruticosa* (Greek sage), *Origanum majorana* (marjoram), and *Mentha spicata* (spearmint) were among the recorded medicinal plant species. Interestingly, the same plants’ concoction resulted in the prevention and cure of the common cold and influenza. At the same time, the exploration of the antioxidant activity of their extracts (without isolation and identification of the active compounds) was directed to detect a possible underlying mechanism of biological action.

### 2.3. Prospecting

#### 2.3.1. Bioprospecting

The definition of bioprospecting involves the systematic search for genes, natural compounds, designs, and whole organisms in wildlife with potential for product development [72]. For the needs of this work, however, this term is restricted to the exploration, utilization, and exploitation of the plants’ biological diversity, either within the context of traditional medicinal knowledge or outside it. The collaboration between Merck Co. and Costa Rica’s National Institute of Biodiversity is a much-cited example of successful conventional bioprospecting, having identified novel compounds from fungi, such as arundifungin and durhamycin A, a novel antifungal compound and potent inhibitor of HIV Tat transactivation, respectively. Nevertheless, many other bioprospecting programs have not been as successful, impelling the exploration away from the assistance and primary knowledge of traditional healers [73].

Undoubtedly, the success of a commercial target necessitates a follow up across the translational chain, beyond the quality of the research product itself. Therefore, drug discovery and development is a long, costly, and high-risk process that takes over 10–15 years, while the attainment rate in translation, from R&D (preclinic) to the clinic stage, is less than 1% [70,74]. Harrison reported that in 174 drug development failures for the period 2013–2015, the majority of cases were due to a lack of either efficacy (52%) or safety (24%, including an insufficient therapeutic index). Strategic (15%), commercial (6%), and operational (3%) reasons were cited for the remainder of the failures [75]. Therefore, several efforts took place in order to facilitate future bioprospecting. Specific attention was given to the cross-cultural corroboration of medicinal usage of natural compounds or plants to guide bioprospecting. Roersch [76], in his ethnomedicinal review of *Piper umbellatum*, a species found in the Americas, Africa, and Asia, has recorded its use in 24 countries, stating that those with consensus across different cultures are more likely to be supported with scientific evidence and should be prioritized in pharmacological studies [76]. In this line, large-scale cross-cultural comparisons of ethnomedicinal floras were conducted, incorporating new phylogenetic and statistical methodologies [77,78,79,80], with promising results.

Based on the efficacy of combinations of aromatic plants reported previously [71] for the prevention of common cold and influenza, we prepared a combination of *Thymbra capitata* (thyme)*, Origanum dictamnus* (Creta dittany), and *Salvia fruticosa* (Greek sage). The preparation was found to be safe in animals and humans [81] and efficient in vitro against a variety of upper respiratory tract viruses, including strains of influenza [82]. Interestingly, in vitro assays revealed that the preparation inhibited the nuclear translocation of the viral nucleoprotein, providing evidence for a specific mechanism of action [82] (see also the next paragraph). A clinical trial performed with the preparation reported its potency in treating influenza infections [83], and a post market analysis, after commercialization of the product as a dietary supplement, confirmed these data [84].

Performing detailed computational studies with ingredients of our preparation and taking into account our results of potentially active compounds [85] (see Section 2.4), we tested our preparation in vitro against SARS-CoV-2—infected cells. We found (Figure 2A) that it can promote the survival of cells after infection, reducing viral replication, both after pre- or coincubation with CAPeo, while, in a proof-of-principle study, in ambulatory COVID-19 patients, it induces a rapid elimination of disease-related symptoms (Figure 2B). A clinical trial is currently in progress.

#### 2.3.2. Mass Bioprospecting

Large-scale explorations, largely guided by the so-called “biodiversity” or “random” collection approach, where ethnobotanical or ethnopharmacological information plays a minimal or no role, are known as “mass bioprospecting” [27]. The most cited mass bioprospecting example concerns the efforts of the United States National Cancer Institute (NCI) in searching for plant-derived anticancer agents. About 114,000 extracts from an estimated 35,000 plant samples (representing 12,000–13,000 species), collected mostly from temperate regions of the world, had been screened against a number of tumor systems between 1960 and 1982 [27,86,87]. Despite the random collection approach and high-throughput methods of screening, mass bioprospecting of the NCI was also characterized by the fundamental feature of traditional pharmacology, which is the use of herbal formulae as the typical treatment. The herbal formula contains hundreds of chemical compounds. This complexity makes this approach complicated, time consuming, and challenging in understanding the mechanisms of action and bioactive ingredients. In 1983, the NCI’s mass bioprospecting effort was extended through the establishment of a National Cooperative Drug Discovery Group (NCDDG) program by the Developmental Therapeutics Program (DTP), Division of Cancer Treatment and Diagnosis (DCT).

The high failure rates, both in bioprospecting and mass bioprospecting, have raised the question of whether certain aspects of drug development are overlooked. Despite unprecedented investment in drug development and the major advances in many of the scientific and technological inputs into drug research and development (R&D), the number of new drugs approved by the US Food and Drug Administration (FDA) for the period 1950 to 2010 was low [86,87]. The number of new drugs approved per billion US dollars spent on R&D in this period has halved roughly every 9 years since 1950, falling around 80-fold in inflation-adjusted terms [86]. Therefore, a number of elements have been advanced, such as whether the new drug output may simply reflect the limitations of the current R&D model [87]. For example, given that most of the costs of new drug development are related to the costs of failed projects, the idea that high-affinity binding to a single biological target linked to a disease will lead to medical benefit in humans should be reevaluated. This has led to the conclusion that if the causal link between single targets and disease states is weaker than commonly thought, or if drugs rarely act on a single target, then the molecules that have been delivered by this research strategy into clinical development may not necessarily be more likely to succeed than those in earlier periods [86]. Finally, an interesting hypothesis has been advanced by Firn [88]. The author suggests that, even in the case of a positive hit in bioprospecting, the cost of synthesis of identified chemicals is high, given the complexity of natural molecules, an element that discourages pharmaceutical companies from investing in such molecules, investing in R&D, and subsequently decreasing the interest of investors in ethnopharmacology. In contrast, living organisms, having the correct enzymatic system, are capable of an efficient synthesis of such complex molecules. This situation is actually reversing, in view of the progress in molecular biology and ex vivo synthetic capabilities, increasing again the interest in natural product research.

### 2.4. Computational Chemistry

Computational drug discovery has over the past few decades become very relevant mainly due to the reduced risks, time, cost-effectiveness, and resources as compared with the traditional experimental approaches [89]. The substantial increase in computational power, the development and implementation of artificial intelligence methods, and the availability of huge freely available data collections were the main reasons for the development of computational methods with a special impact on novel drug compound identification. The development of novel powerful analytical techniques, as described above, has permitted the implementation of computational methods in the field of natural product derivatives (see [90] for a critical review and presentation of available resources). Computational methods include, among others: (1) the 3D resolution of the conformation of a large number of noncrystallized proteins [91,92], using modern artificial intelligence methods, with AlphaFold and AlphaFold2 being the more successful; (2) molecular docking developments, permitting the fully flexible association of (druggable) compounds to their putative targets (see [93] for a discussion). This was made possible with the recent increase in computational power. Indeed, older solutions considered the macromolecular target of drugs as a rigid molecule and tried to associate a rigid or flexible micromolecule/drug/natural product at a given binding pocket. However, it is known that the conformation of the macromolecule is equally modified by the binding of the ligand. With the increase in computational power, fully flexible solutions have been developed, in which all atoms of the macromolecule are also modified by the binding, providing a more accurate determination of the binding affinity [93]. These methods have been enhanced by molecular dynamics simulations, permitting the calculation of the movements of all atoms in the micromolecule for short (fsec) to very long (μsec) periods of time and the resulting conformational poses of the complex macromolecule–ligand, along with enhanced sampling techniques and elaborate methods of analysis, which have allowed unprecedented insight into complex phenomena in biology at extreme efficiency and accuracy. The combination of these methods, especially for the conformational changes of proteins, has permitted the simultaneous detection of movements and conformational states of thousands or millions of atoms and molecules in a given structure [94,95,96]; (3) the development of solutions mimicking the molecular mechanisms triggered by the activation of the (druggable) micromolecule to its (protein) target (see [85,97,98] for a discussion and a concrete example); (4) the development of “network pharmacology” methods [26,99,100,101,102,103,104,105], in which the exploitation of experimental and/or bibliographic evidence of signaling pathways and specific effects is explored, with increasingly sophisticated methods (see for example [99,106]), lead to a prediction of potential effects of a given substance, including natural products (see the thematic issue in ref. [100]) for a recent discussion on the subject); (5) the development of QSAR (quantitative structure–activity relationship) methods, in which, through sophisticated statistical methods, “active” parts of the ligand/drug/natural product, at a given conformation are extracted and predictions about a large number of putative or potential novel ligands can be drawn (see [107,108,109] for a critical analysis, pitfalls, and solutions of this methodology); (6) the development of bioinformatic methods for the prediction of absorption–distribution–metabolism and excretion of druggable molecules (ADMET), integrating artificial intelligence methods (see [110,111,112,113] for concrete examples and solutions).

With the help of computational chemistry, thousands of molecules have been evaluated for potential efficacy and safety at a small cost in a very short interval of time [114], overcoming the limitations of the experimental approach, helped by progress in artificial intelligence [115]. The lead compounds should have high-affinity prospective binding and specificity for a target associated with a disease and favorable pharmacodynamic and pharmacokinetic properties [22]. Of course, being a cheaper and less time-consuming process, when compared to experimental high-throughput screening, computational chemistry and pharmacology are expected to increase the output of the drug development process. Indeed, the median number of new molecules and biological license application approvals from 2010 to 2019 has increased by 60%, compared to the prior decade. However, the productivity of the pharmaceutical industry may be also linked to regulatory incentives, such as breakthrough therapy, fast-track designation, and Orphan Drug and GAIN acts, through which a large proportion of new drugs have been approved. As a concrete example, increased investments in basic neuroscience research by regulatory research authorities, such as the NIH or the European Union, have recently been linked to an increase in CNS startup investments [116].

It is estimated that computational chemistry has increased the quality of filtering and selection and improved the filtering efficiency by several orders of magnitude but without increasing the output substantially, mainly due to the bottlenecks of the R&D chain, as the necessity for experimental and clinical trials remains [115]. Indeed, despite the increasing number of successful applications in prospective computational chemistry, the computational methods are still limited to reliably predicting biological activity from chemical structure [117].

The discrepancies between preclinical research, with the use of computational chemical methods and clinical results, need a thorough evaluation, as only a small part of the computationally suggested compounds in the literature are experimentally tested, and negative results are not published. Moreover, many computational screening applications in industrial R&D departments are kept confidential, and compounds of interest are not disclosed, before a successful intellectual property filing. The reverse is also true, as disclosing candidate active compounds for drug development without patent protection discourages possible investments from the pharmaceutical industry (see [118] for a recent overview and discussion). Therefore, integrating computational and preclinical experimental screening (see [119] for a discussion) is necessary for a successful prospective screening, both for the strong patent protection and for further efficacy improvement of clinical trials, influencing the output of the drug production process. However, a number of successes have been acknowledged, with a number of novel drugs already approved by regulatory authorities and being on the market (Table 2), while an extensive list of other, potential druggable natural product candidates is under investigation [120,121]. The interested reader should consult Issues 7 and 8 of the journal *Drug Discovery Today* (2022) for a series of articles on the subject).

Our group has extensive and long-standing experience in exploring in silico, in vitro, and in vivo the effect of natural compounds as potential pharmacological agents. Since 2000, we have provided evidence that red wine polyphenols act as antiproliferative agents in breast and prostate cancer cells [127,128]. Later on, we reported that proanthocyanidins [129,130], and especially proanthocyanidin B2, was the most potent compound, acting on a cell membrane androgen receptor [131], which was later characterized as the receptor binding oxo-eikosanoids (OXER1) [132], on which testosterone and polyphenols act as antagonists, modifying cAMP production [132], actin cytoskeleton [130,131,132], and intracellular Ca^2+^ [133,134]. Proanthocyanidins were repeating the effects of testosterone, modifying actin cytoskeleton and cAMP production in vitro [131,133], and inducing apoptosis in vitro and the regression of tumors in vivo in BalbC^−/−^ mice breast and prostate xenografts [131]. Using testosterone and polyphenols as baits, we explored the ZINC database of natural products, taking advantage of a developed bioinformatic resource, permitting the classification of compounds as agonists or antagonists, through the simulation of G_α_ protein binding to the ligand-bound G-protein coupled receptors [98]. Using quantitative structure–activity relationships (QSAR), followed by in silico binding simulations and extensive in vitro validation, we identified ZINC15957997 as a specific OXER1 G_αi_ antagonist [135], ZINC8589130 as a specific OXER1 G_βγ_ antagonist and ZINC4017374 as an OXER1 pan-G-protein antagonist (Panagiotopoulos et al., in preparation). In vivo experiments with these compounds in BalbC^−/−^ mice xenografts of prostate cancer are programmed.

In another field, we interrogated the natural products database for potential inhibitors of SARS-CoV-2 infections. We retained fortunellin as an allosteric inhibitor of the main viral protease dimerization (an absolute requirement for its action). This identification was made through in silico binding simulations, followed by molecular dynamics in long simulation times (10 μs) and validated in vitro, in SARS-CoV-2–infected Vero cells [97]. Fortunellin is found in kumquat, while its homolog apiin was found in parsley and celery. Finally, we identified p-cymene, a main constituent of the essential oil mixture described in Section 2.3.1, as a potent anti-influenza and anti-SARS-CoV-2 agent, acting in viral nucleoprotein and nucleocapsin viral proteins, respectively. Initial in silico studies have also been verified in vitro in infected cells [85].

## 3. Conclusions

The development of new drugs from herbal plant ingredients has been the basic agenda in the R&D of the drug industry for many decades. Either random or knowledge-based selection of plants could potentially reveal valuable compounds for the drug industry. In this brief review, we provided evidence on the evolution of concepts and processes used in the field of natural products’ use for drug development, sustained by our experience in the field. We provided evidence about the development of concepts leading from the ethnopharmacological or ethnobotanical observations of the beneficial effects of plant concoctions or essential oils and end up with the isolation and characterization of specific compound(s) for use as drugs or food supplements. Many emblematic drugs (Taxol [3,4,5], Vinblastine [6,7], quinine [8,9], and artemisinin [10]) have their origin in traditional medicine and ethnopharmacology, although later, they have been synthesized and re-evaluated with modern analytical and pharmacological methods.

However, the large expansion of scientific analytical and detection methods, the tremendous increase in computational power, and the construction of large public natural compounds libraries have led to a paradigm shift in the pharmacological use of plants and plant extracts. Analysis, isolation, and combinatorial use of single or multiple “active” ingredients have been used for the treatment of specific diseases and conditions (see [34] for a discussion and a concrete example). The development of biology and clinical and experimental medicine has facilitated this transition by providing specific target molecules or molecular pathways in which active plant ingredients have a specific effect. This bottom-up approach has led to the detection of specific compounds, available for experimental testing, and redefining the flow direction of traditional natural product use in pharmacology. Indeed, in spite of the traditional plant initiation point to the isolation and exploitation of (an) active compound(s), now, the reverse is also possible. Indeed, plant selection, in many cases, may result after the detection, characterization, and biological evaluation of a specific compound. Ranking compounds according to pharmaceutical relevance has been made possible due to their ability to predict the putative binding affinities between small molecules and biological counteractors, with potential therapeutic traits. Computational tools have helped to define and elaborate the strength of interaction between ligands and targets and have been instrumental in the identification of lead molecules from databases. However, despite the high expectations that computational chemistry would translate into increased production of new drugs, the result showed limited success. As we stressed and showed with concreter paradigms, the computational identification and the combinatorial in silico identification of compounds is only the first step in successful drug discovery. This should be supported with at least an in vitro validation of results. The whole process of drug development (in silico, in vitro, in vivo, clinical trials) should be respected for the novel or repurposed use of natural compounds for successful drug development, an element that maintains a high cost and sustained effort. This is supported by the explosion of publications during the current COVID-19 pandemic, in which, in spite of the in silico prediction of active compounds or the repurposing of existing drugs (with the notable examples of colchicine and nicotine), few novel drugs have been advanced and made available.

It is now clear that beyond the one-dimensional explanations such as the continuous need for further improvements in prediction algorithms, all the R&D operations, which are the core aspect of drug discovery and development, are extensively affected and controlled by a broad socioeconomic context that has been defined as a “pharmaceutical ecosystem”. The pharmaceutical ecosystem refers to the interdependent relationships among levels of interacting stakeholder networks, in connection with processes, tools, and infrastructures that are controlled by policies, laws, and opinions [136], which influence the production rate of new medicines. In this context, the accelerated research of natural compounds based on ethnopharmacological observations keeps a prominent role.

## 4. Patents

This work mentions the following patents (and those derived from them) in which one or multiple authors are named as inventors: US2008227853A1, WO2004006966A1, WO2011045557A1, WO2007123682A3, WO2021160768A1, and WO2012038694A1.

## Figures and Tables

**Figure 1 molecules-27-04060-f001:**
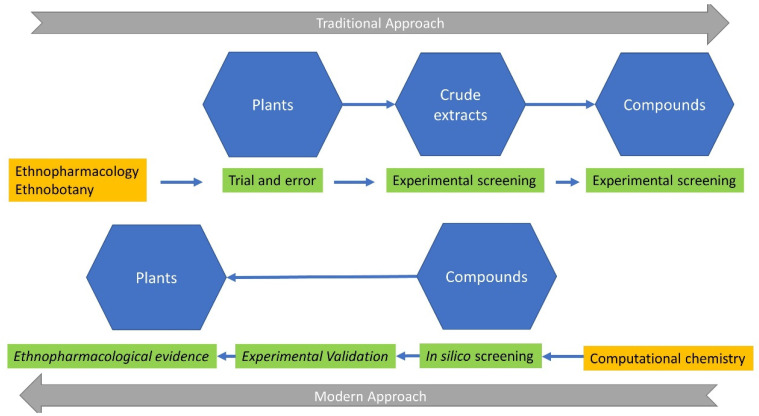
Scientific disciplines (orange color), methodologies (green color), and compartments (blue color) from traditional to modern approaches in ethnopharmacology. Arrows denote the transition from the trial-and-error methodologies of the indigenous people to the in silico screening of computational chemistry. It is of note, as discussed in the text, that modern ethnopharmacology incorporates the computational and experimental validation of active natural compounds, prior to the detection of ethnopharmacological evidence. See text for further details.

**Figure 2 molecules-27-04060-f002:**
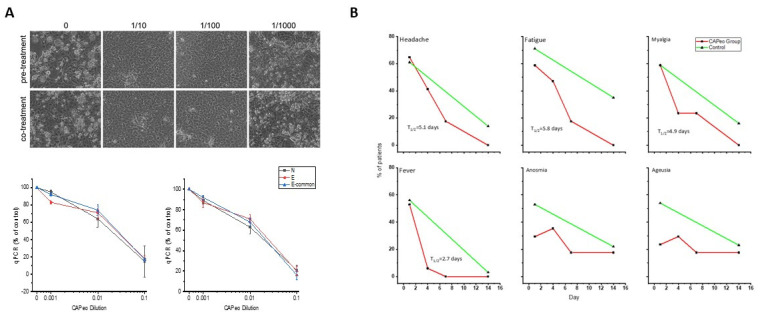
Effect of a mixture of three aromatic plants ((thyme, Greek sage, and Cretan dittany) in SARS-CoV-2 infection. (**A**). In vitro testing. Left panel: light microscopy photographs of CPE in control (0, DMSO) and SARS-CoV-2–infected Vero cells (strain B.1, 0.1 m.o.i), pretreated or cotreated with different concentrations of CAPeo, in DMSO. Lower panel: Curves representing relative abundance (% of untreated control) of SARS-CoV-2 RNA after pretreatment (left curves) or cotreatment (right curves) with different concentrations of CAPeo, using real-time quantitative RT-PCR, targeting N and E regions of SARS-CoV-2 genome and E-common region shared by SARS-CoV and SARS-CoV-2 viruses. Values are shown as mean ± SD of three separate measurements. No significant differences in pretreated or cotreated cells were found. In both cases, CAPeo was efficient in concentrations almost 100 times lower than those administered per os and compatible with the estimated circulating concentration of the product. (**B**). Evolution of selected symptoms in our CAPeo-treated group (red curves), in a proof-of-principle trial (NCT04705753). T_1/2_ for the resolution of symptoms was calculated with a logistic regression fit. For comparison, the frequency of symptoms in the reference populations is also presented (green curves).

**Table 1 molecules-27-04060-t001:** Indicative studies of the research shift from identifying and recording the medicinal plant species used in traditional medicine to the evaluation of specific properties or treatment effects of crude plant extracts, or particular naturally derived chemical substances.

Author	Year	Title	Reference
*Ethnopharmacological/ethnobotanical studies on traditional medicine*			
Ahmet Sargin, S.	2015	Ethnobotanical survey of medicinal plants in Bozyazi district of Mersin, Turkey.	[30]
Alzweiri et al.	2011	Ethnopharmacological survey of medicinal herbs in Jordan, the Northern Badia region.	[43]
Karousou et al.	2011	The herbal market of Cyprus: traditional links and cultural exchanges.	[44]
Al-Qura’n, S.	2009	Ethnopharmacological survey of wild medicinal plants in Showbak, Jordan.	[45]
Hudaib et al.	2008	Ethnopharmacological survey of medicinal plants in Jordan, Mujib Nature Reserve and surrounding area.	[46]
Lardos, A.	2006	The botanical materia medica of the Iatrosophikon--a collection of prescriptions from a monastery in Cyprus.	[47]
Said et al.	2002	Ethnopharmacological survey of medicinal herbs in Israel, the Golan Heights and the West Bank region.	[48]
Lev et al.	2000	Ethnopharmacological survey of traditional drugs sold in Israel at the end of the 20th century.	[49]
Ali-Shtayeh et al.	1998	Antimicrobial activity of 20 plants used in folkloric medicine in the Palestinian area.	[50]
Vázquez et al.	1997	Medicinal plants used in the Barros Area, Badajoz Province (Spain).	[51]
Honda et al.	1996	Traditional medicine in Turkey VI. Folk medicine in West Anatolia: Afyon, Kütahya, Denizli, Muğla, Aydin provinces.	[52]
Al-Khalil, S.	1995	A Survey of Plants Used in Jordanian Traditional Medicine.	[53]
Dafni et al.	1984	Ethnobotanical survey of medicinal plants in northern Israel.	[54]
*Evaluation of specific properties or treatment effects of crude plant extracts*			
Qnais et al.	2007	Antidiarrheal Activity of the Aqueous Extract of *Punica granatum*. (Pomegranate) Peels.	[55]
Hage-Sleiman et al.	2011	Pharmacological evaluation of aqueous extract of *Althaea officinalis* flower grown in Lebanon.	[56]
Hsu et al.	2007	Antioxidant activity of extract from *Polygonum cuspidatum*.	[57]
Hsu	2006	Antioxidant activity of extract from *Polygonum aviculare* L.	[58]
Perianayagam et al.	2006	Anti-inflammatory activity of *Trichodesma indicum* root extract in experimental animals.	[59]
Garrido et al.	2004	In vivo and in vitro anti-inflammatory activity of *Mangifera indica* L. extract (VIMANG).	[60]
Rajeshkumar et al.	2002	Antitumour and anticarcinogenic activity of *Phyllanthus amarus* extract.	[61]
Olajide et al.	2000	Studies on the anti-inflammatory and related pharmacological properties of the aqueous extract of *Bridelia ferruginea* stem bark.	[62]
Bhakta et al.	1998	Studies on Antitussive Activity of *Cassia fistula* (Leguminosae) Leaf Extract.	[63]
*Evaluation of specific properties or treatment effects of particular naturally derived chemical compounds*			
Górniak et al.	2019	Comprehensive review of antimicrobial activities of plant flavonoids.	[64]
Adamski et al.	2020	Biological Activities of Alkaloids: From Toxicology to Pharmacology.	[65]
Pizzi	2021	Tannins medical/pharmacological and related applications: A critical review.	[66]
Metwaly et al.	2019	Black Ginseng and Its Saponins: Preparation, Phytochemistry and Pharmacological Effects.	[67]
Lu, et al.	2002	Polyphenolics of Salvia—a review.	[68]
Yang et al.	2020	Advances in Pharmacological Activities of Terpenoids	[69]

**Table 2 molecules-27-04060-t002:** Recent successful stories of computational drug discovery approved by FDA or in a clinical trial. For an extensive Table of commercial drugs that made use of computer-aided drug design during the discovery process see [121], while for a detailed list of proteins and phytocompounds for computational docking along with therapeutic potential see [120].

Drug	Description	Reference
Crizotinib	Crizotinib has been considered as a selective and potent cMet/ALK dual inhibitor, which was approved by FDA in 2011. Crizotinib has demonstrated remarkable clinical efficacy on c-MET gene amplification against lung cancer, lymphoma, and esophageal cancers.	[122,123]
Axitinib	Axitinib was approved by the FDA as a new therapy for advanced renal cell carcinoma to treat VEHG. Axitinib was developed with a structure-based drug design strategy and served as an inhibitor by binding to the VEGF kinase domain in the DFG-out conformation	[123,124,125]
Grazoprevir	Grazoprevir is an NS3/4a protease inhibitor developed for the treatment of hepatitis C virus (HCV). Grazoprevir successfully passed clinical trials and was approved in 2016 against hepatitis C.	[126]
Betrixaban	Betrixaban was FDA approved in 2017 for the prophylaxis of venous thromboembolism (VTE) in adult patients hospitalized for an acute medical illness, who are at risk for thromboemboliccomplications due to moderate or severe restricted mobility.	[126]
Vaborbactam	Vaborbactam was FDA approved in 2017 for complicated urinary tract infections (cUTI), including a type of kidney infection, and pyelonephritis, caused by specific bacteria.	[126]
Luminespib (NVP-AUY922)	HSP90 has become a promising target for cancer treatment. Luminespib (NVP-AUY922) has been proved to be a strong HSP90 inhibitor which is now in clinical trials.	[123]

## Data Availability

All data are included in the text and the Figures.

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
