# Peer review of "From Traditional Ethnopharmacology to Modern Natural Drug Discovery: A Methodology Discussion and Specific Examples"

_molecules, 2022, doi:10.3390/molecules27134060_

Round 1
Reviewer 1 Report
This manuscript reviewed the traditional and modern drug discovery of natural medicines. The topic is interesting, but the content is not comprehensive and sufficient for publication based on the current version.
1. The language needs to be extensively edited. There are numerous grammatic errors.
2. The authors reviewed four sessions: Traditional ethnopharmacology and ethnobotany, Pharmacological testing, Prospecting, and Computational chemistry. However, none of them has been comprehensively discussed and reviewed.
(1) For the first part, only the current use of herbs in various countries has been briefly mentioned, while the recent discovery/development is not summarized. The authors listed their contribution to the safety of herbal medicine, which does not fit for this session.
(2) The title of “Pharmacological testing” is a very broad topic and the author did cite a lot of literature. However, there is no discussion or summary except for the citations only.
(3) For the session of “Prospecting”, the authors listed the risk of new drug development and their own studies on herbal medicines. However, there is no information on “Prospecting”. In addition, there is no review of cases or reports of “Mass bioprospecting”, except for some general numbers. In general, the main session of this review is not informative.
3. Since the topic is too broad, the authors may want to restrict their review to certain areas, such as COVID-related, or certain diseases, or particular research areas, etc.
4. The figure is not clear, which needs to be replaced.
5. Tables of summary should be used, since many literatures are listed.
Author Response
- The language needs to be extensively edited. There are numerous grammatic errors.
The language was polished, and a number of errors have been corrected.
- The authors reviewed four sessions: Traditional ethnopharmacology and ethnobotany, Pharmacological testing, Prospecting, and Computational chemistry. However, none of them has been comprehensively discussed and reviewed.
We agree with the reviewer that the fields addressed are very extensive. Therefore a systematic review of all these scientific disciplines/fields is not possible in the context of a single review article. Our scope here was to outline the fields, in order to provide the flow of ideas, leading from ethnopharmacological observation to drug discovery and not to provide an in extenso review of the field. This is now explicitly described in L129-142, supported by Figure 1.
(1) For the first part, only the current use of herbs in various countries has been briefly mentioned, while the recent discovery/development is not summarized. The authors listed their contribution to the safety of herbal medicine, which does not fit for this session.
In line with the reviewer's comment, we have added recent discoveries in the Introduction, supported by corresponding literatu200-216).
(2) The title of “Pharmacological testing” is a very broad topic and the author did cite a lot of literature. However, there is no discussion or summary except for the citations only.
This part has been expanded, according to the reviewer's comment and supported by Table 1, as requested. We further provide a discussion on the potential pitfalls and delays in drug discovery.
(3) For the session of “Prospecting”, the authors listed the risk of new drug development and their own studies on herbal medicines. However, there is no information on “Prospecting”. In addition, there is no review of cases or reports of “Mass bioprospecting”, except for some general numbers. In general, the main session of this review is not informative.
In accordance to the reviewer comment, we have substantially increased the length and provided additional information and critical discussion in this part of the manuscript.
- Since the topic is too broad, the authors may want to restrict their review to certain areas, such as COVID-related, or certain diseases, or particular research areas, etc.
Indeed, the topic is too broad, therefore we have tried to do clear that we did not do a detailed review of all these areas. Instead, we tried to see what changed the approach from ethnopharmacology to drug discovery, in the traditional way (observation to drug) and the modern approach (in silico evaluation, compound discovery and back to the natural origin of the identified substance). We hope that this is now clear in the manuscript.
- The figure is not clear, which needs to be replaced.
We have provided additional information on the meaning of Figure 1, and increased the resolution of Figure 2.
- Tables of summary should be used, since many literatures are listed.
Accoeding to the reviewer's suggestion, we have incorporated now Tables 1 and 2, detailing the cited literature.
Reviewer 2 Report
Authors present a manuscript that is a very short review (or a short communication), with a focus on their own work. In total, it is not comprehensive enough for publication. More examples should be added, if available, or it should be supported more exactly, why is this focus.
On Fig 2A this referee could not find significant differences between pre-treatment and co-treatment. Is this right, is that rationalized?
There are a couple of similar "opinion" communications available in the literature.
https://academic.oup.com/jpp/article/53/4/425/6149869?login=false
https://www.sciencedirect.com/science/article/pii/S1674638420300204
https://www.ncbi.nlm.nih.gov/pmc/articles/PMC3151376/
https://www.ncbi.nlm.nih.gov/pmc/articles/PMC3901206/
https://www.jstor.org/stable/3434847
https://www.frontiersin.org/articles/10.3389/fphar.2010.00008/full
It is suggested to check whether these could be or should be included/mentioned.
In many cases authors suggest to check a literature example and find the discussion/examples there. It is suggested to include more from these papers in order to make the manuscript a bit longer and more detailed, comprehensive.
"Concrete" in my opininon is not the best word as it has a secondary meaning as well. Would it be possible to find a synonym?
Author Response
1. Authors present a manuscript that is a very short review (or a short communication), with a focus on their own work. In total, it is not comprehensive enough for publication. More examples should be added, if available, or it should be supported more exactly, why is this focus.
We thank the reviewer for the positive appreciation of our work. In the current manuscript, we have substantially extended the length and included information and cited literature. However, as pointed in L129-140, the scope of this communication was not to provide an in depth review of all fields leading from ethnopharmacological observation to drug discovery, but to present the traditional and modern approaches of natural products pharmaceutical discovery, a very rapidly evolving field.
2. On Fig 2A this referee could not find significant differences between pre-treatment and co-treatment. Is this right, is that rationalized?
This is correct! We now include a comment in L307-308.
3. There are a couple of similar "opinion" communications available in the literature.
https://academic.oup.com/jpp/article/53/4/425/6149869?login=false
https://www.sciencedirect.com/science/article/pii/S1674638420300204
https://www.ncbi.nlm.nih.gov/pmc/articles/PMC3151376/
https://www.ncbi.nlm.nih.gov/pmc/articles/PMC3901206/
https://www.jstor.org/stable/3434847
https://www.frontiersin.org/articles/10.3389/fphar.2010.00008/full
It is suggested to check whether these could be or should be included/mentioned.
We thank the reviewer for this suggestion. All the relevant information pointed-out have been now incorporated into the text.
4. In many cases authors suggest to check a literature example and find the discussion/examples there. It is suggested to include more from these papers in order to make the manuscript a bit longer and more detailed, comprehensive.
According to the reviewer's suggestion, we have now incorporated relevant information and literature in order to further clarify the topics discussed in this review.
5. "Concrete" in my opininon is not the best word as it has a secondary meaning as well. Would it be possible to find a synonym?
Real? Actual? Specific? Discrete?
We have replaced "concrete" by "specific, according to the reviewer's suggestion.
Reviewer 3 Report
This review article discusses the evolution of ideas, from the traditional ethnopharmacology to in silico drug discovery, applied to natural products. The authors also present our contribution in this evolution by discussing discrete examples of novel drug discovery. Before recommending this article for publication, there are some shortcomings for that should be resolve.
General comments
Overall the experiment is well designed and presented in a good way but English of the whole manuscript should be revised. Long sentences should be avoid.
Abstract
This section is well written but conclusion should be containing one sentence on future recommendation.
There are very long sentences which must be revised and clarify for readers understanding.
Also discuss some main findings and conclusion based on the review.
Introduction
Line 36-38 poor English must be revise.
Lie 50-51 provide the plant names as well from which these drugs are derived and also mention diseases for which they are used.
Line 54-61 must be cited.
Line 77-81 must be cited the following references may be helpful.
DOI: http://dx.doi.org/10.30848/PJB2022-3(19), https://doi.org/10.1016/j.jep.2021.114515,
The authors should mention some modern computational techniques in introduction. Briefly.
Section 2.1. Line 137 the author states that “In contrast, western societies have mostly lost their traditional healing practice”. However, there are many institutions and organizations in working and relying on traditional medicines in the western world. The statement is not authentic.
Section 2.2. first provide introduction or definition and mechanism of pharmacological testing. After that review of literature.
Present a flow chart or visual graphics for the methods of bioprospecting and computational chemistry.
The authors should mention some prominent drugs or compounds, its uses against diseases, plants sources etc. obtained as result of computational chemistry or bioprospecting. A table would be better.
This manuscript can be accepted after recommended revision.
Author Response
- This review article discusses the evolution of ideas, from the traditional ethnopharmacology to in silico drug discovery, applied to natural products. The authors also present our contribution in this evolution by discussing discrete examples of novel drug discovery. Before recommending this article for publication, there are some shortcomings for that should be resolve
We thank the reviewer for the positive appreciation of our work. Please, refer below for the specific elements pointed-out.
2. General comments
2.1. Overall the experiment is well designed and presented in a good way but English of the whole manuscript should be revised. Long sentences should be avoid.
In this version, we have significantly polished the language of the manuscript.
2.2. Abstract
This section is well written but conclusion should be containing one sentence on future recommendation.
There are very long sentences which must be revised and clarify for readers understanding.
Also discuss some main findings and conclusion based on the review.
We have tried to simplify the language, according to the suggestion of the reviewer. We have added a summary of our work in L24-33 of the Abstract, as requested..
2.3. Introduction
Line 36-38 poor English must be revise.
Lie 50-51 provide the plant names as well from which these drugs are derived and also mention diseases for which they are used.
Line 54-61 must be cited.
Line 77-81 must be cited the following references may be helpful.
DOI: http://dx.doi.org/10.30848/PJB2022-3(19), https://doi.org/10.1016/j.jep.2021.114515,
The authors should mention some modern computational techniques in introduction.
All the suggestions of the reviewer were taken into account and the text was revised accordingly.
2.4. Section 2.1. Line 137 the author states that “In contrast, western societies have mostly lost their traditional healing practice”. However, there are many institutions and organizations in working and relying on traditional medicines in the western world. The statement is not authentic.
We thank the reviewer for poining out this poor sentence. Here, we have expanding this section, and further clarified the issue. We point out the regulations in place for the general use of phytotherapy in the eastern and western societies and the existing regulatory mechanisms (L162-186), while pointing also the place of ethnopharmacology as a valuable source for drug development and as a vehicle for the preservation of indigenous knowledge (L187-192).
2.5. Section 2.2. first provide introduction or definition and mechanism of pharmacological testing. After that review of literature.
The text has been expanded, according to the reviewer's suggestion and Table 1, included in this section, helps to better understand the subject. However, as we point out in the Introduction, the scope of this work was to set in place specific mechanisms and flow of ideas rather, than to provide an extensive review of each step leading from ethnopharmacology to drug discovery.
2.6. Present a flow chart or visual graphics for the methods of bioprospecting and computational chemistry.
We have opted not to include an additional graph, for this section, as the relevant information is included in the lower part of Figure 1. However, we have expanded the description of this approach in L99-122 and L131-142 of the manuscript.
2.7. The authors should mention some prominent drugs or compounds, its uses against diseases, plants sources etc. obtained as result of computational chemistry or bioprospecting. A table would be better.
The text has been expanded and a relevant Table 2 was included, according to the reviewer's suggestion.
Round 2
Reviewer 1 Report
The revision appears to be much improved. No further comments.
Reviewer 2 Report
Authors modified the manuscript accordingly to the comments and questions, and thus a significant change has been made. Now in the opinion of this referee the manuscript could be accepted for publication.
Reviewer 3 Report
The revised version provided by authors are ready for acceptance.